# ESCNet: Entity-enhanced and Stance Checking Network for Multi-modal Fact-Checking

Submission Id: 779

## ABSTRACT

Recently, misinformation incorporating both texts and images has been disseminated more effectively than those containing text alone on social media, raising significant concerns for multi-modal fact-checking. Existing research makes contributions to multi-modal feature extraction and interaction, but fails to fully enhance the valuable semantic representations or excavate the intricate entity information. Besides, existing multi-modal fact-checking datasets are primarily focused on English and merely concentrate on a single type of misinformation, thereby neglecting a comprehensive summary and coverage of various types of misinformation. Taking these factors into account, we construct the first large-scale Chinese Multi-modal Fact-Checking (CMFC) dataset which encompasses 46,000 claims. The CMFC covers all types of misinformation for fact-checking and is divided into two sub-datasets, Collected Chinese Multi-modal Fact-Checking (CCMF) and Synthetic Chinese Multi-modal Fact-Checking (SCMF). To establish baseline performance, we propose a novel Entity-enhanced and Stance Checking Network (ESCNet), which includes Multi-modal Feature Extraction Module, Stance Transformer, and Entity-enhanced Encoder. The ESCNet jointly models stance semantic reasoning features and knowledge-enhanced entity pair features, in order to simultaneously learn effective semantic-level and knowledge-level claim representations. Our work offers the first step and establishes a benchmark for evidence-based, multi-type, multi-modal fact-checking, and significantly outperforms previous baseline models.

## CCS CONCEPTS

• **Information systems → Social networks**; **Multimedia information systems**.

## KEYWORDS

Multi-modal fact-checking; Datasets; Knowledge graph

## 1 INTRODUCTION

Fact-checking, defined as the process of evaluating the veracity of claims expressed in written or spoken language with the aid of retrieved evidence, has become increasingly critical [17]. Numerous reports suggest that fabrications can lead to the formation of misconceptions about political candidates among citizens, manipulation of stock prices, and threats to public health. Given the influx of new information and the rapidity of its dissemination, manual fact-checking has proven inadequate, emphasizing the need for automated methods to verify claims and encourage the distribution of truthful information on social media platforms [3, 2].

While significant strides have been made in text-based, single-modal fact-checking task, the advent of multimedia technology has created a new challenge. Perpetrators of rumours now frequently exploit both visual and textual content to attract more attention and

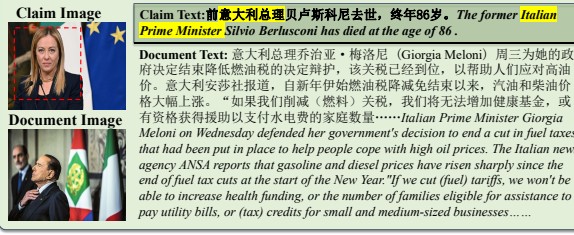

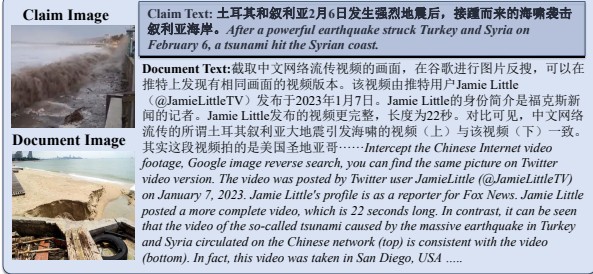

**Figure 1: Two types of misinformation from CMFC (Chinese is translated into English). The source of misinformation within two datasets is different and the entity used to perform a search for irrelevant images is highlighted in yellow.**

expedite dissemination on social media. Compared to single-modal fact-checking, the learning of effective feature representation from heterogeneous multi-modal information poses a greater challenge, rendering multi-modal fact-checking as an intriguing new task [17]. Existing multi-modal misinformation datasets in fact-checking task can generally be divided into two categories: synthetic misinformation [1] and collected misinformation. The difference lies in the source of misinformation within the dataset: Synthetic misinformation refers to the artificially constructed dataset of misinformation created by researchers. Both the text and image of the claim are sourced from pristine news but are deliberately mismatched or partially altered [29]. As shown in Figure 1, the entity 'Italian Prime Minister' used to perform a search for images is highlighted in yellow. A fabricated claim is constructed using the retrieved image of the current Prime Minister, 'Giorgia Meloni', paired with the claim text; While collected misinformation refers to the dataset of misinformation directly gathered from social media platforms. Despite available multi-modal fact-checking datasets primarily focusing on English, they only address one type of misinformation. Moreover, existing multi-modal fact-checking detectors [46, 39, 14, 48] mostly model the basic multi-modal semantic relevance at the feature level, employing concatenate operations [13], or attention mechanisms [16] to capture such coarse semantic correlation and

generate multi-modal representations. Regrettably, the importance of the underlying high-order knowledge and semantic correlation of multimedia content is often overlooked.

To address these challenges, we create the first large-scale Chinese Multi-modal Fact-Checking (CMFC) dataset containing 46,000 claims, which covers two types of misinformation in multi-modal fact-checking task, including Synthetic Chinese Multi-modal Fact-checking (SCMF) and Collected Chinese Multi-modal Fact-checking (CCMF) dataset. The CMFC dataset is of substantial size, with claims and their corresponding evidence documents sourced from a diverse range of platforms and domains.

To establish a baseline performance, we introduce a novel Entity-enhanced and Stance Checking Network (ESCNet) for multi-modal fact-checking task. This network jointly models both stance semantic reasoning features and knowledge-enhanced entity pair features, facilitating the learning of effective semantic-level and knowledge-level claim representations. Specifically, ESCNet consists of Multi-modal Feature Extraction Module, Stance Transformer and Entity-enhanced Encoder (EeE). Given a multi-modal claim and its corresponding retrieved evidence, the Multi-modal Feature Extraction Module initially extracts valuable clues, such as text, images, and entities. Subsequently, we employ three Stance Transformers to simulate the human hierarchical reasoning process, checking the consistency of different types of claims with the evidence. The Stance Transformer first introduces a set of Shared Prototypes as queries, guiding the reconstruction of feature representations of the claim and corresponding evidence, thereby projecting the two features into the common feature space and reducing computation. It then further utilizes a fusion layer to acquire three stance semantic reasoning features. Furthermore, the EeE constructs a cross-modal entity pair set and two uni-modal entity pair sets in the multimedia posts, and designs a knowledge relevance reasoning strategy to find the shortest semantic relevant path between each pair of entities in external knowledge graph. By absorbing all complementary contextual knowledge associated with the entities in this path, the EeE refines knowledge-enhanced distance and entity representations at an elevated knowledge level. It then selects the bottom/top distance entity pairs as the most consistent/inconsistent pairs and the selected entity pairs are further fused by employing a signed attention mechanism to capture consistent and inconsistent knowledge-enhanced entity pair features.

Overall, our main contributions can be summarized as follows: (1) We establish the first large-scale, multi-domain Chinese multi-modal fact-checking dataset, encompassing all types of misinformation in the multi-modal fact-checking task. (2) The proposed ESCNet jointly model both stance semantic reasoning features and knowledge-enhanced entity pair features, facilitating the learning of effective semantic-level and knowledge-level claim representations. (3) We design a Entity-enhanced Encoder with a knowledge-enhanced distance measurement strategy and a signed attention mechanism to capture high-level entity information. (4) Extensive experiments demonstrate the superiority of ESCNet.

## 2 RELATED WORK

**Synthetic Misinformation in Multi-modal Fact-checking.** Synthetic misinformation [1] is one of the most straightforward and

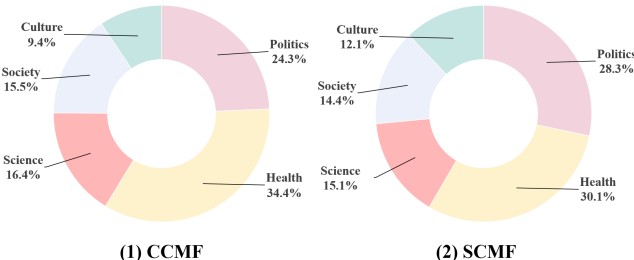

**(1) CCMF**          **(2) SCMF**

**Figure 2: Domain distribution of the two datasets.**

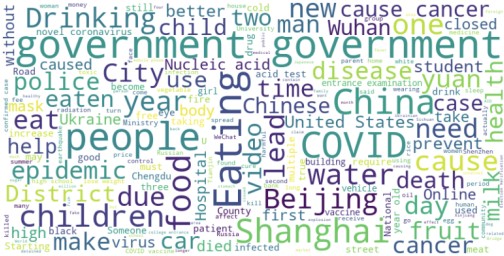

**Figure 3: The word clouds in falsified claims.**

effective strategies that adversaries employ to spread falsified claims [29]. Both the text and image of the statement are sourced from pristine news but are deliberately mismatched or partially altered. Prior work has delved into multi-modal synthetic misinformation: Jaiswal *et al.* [21] directly matched images with titles of other random images, creating its fabricated versions. In contrast, Sabir *et al.* [34] introduced swaps of named entities related to persons, organizations, and locations. Regardless of whether the modifications were made through naive swaps or named entity manipulations, fabricated examples were either overly naive or contained linguistic biases, making them easily detectable even by mere language models [29]. As a result, Luo *et al.* [29] proposed fabricating examples by matching genuine images with genuine texts. They created a large-scale dataset called NewsCLIPpings, which included both pristine and convincingly falsified examples. These fabricated samples might have distorted the context, location, or people in the images, presenting 'out-of-context'[29]. Moreover, Sahar *et al.* [1] compiled a comprehensive NewsCLIPpings dataset by gathering multi-modal evidence from external web sources.

**Collected Misinformation in Multi-modal Fact-checking.** Collected misinformation is directly gathered from the Internet, often propagated as fake news on social media platforms. In recent years, the detection of multi-modal collected misinformation have emerged as a prominent area of focus within the academic community. To address this challenge, a substantial number of researchers dedicated significant efforts to the development and maintenance of datasets that were specifically tailored for this particular field of study [36, 9]. The Factify dataset [31] stood as an early contribution, centered on multi-modal misinformation detection that incorporated both text and image. It consisted of a vast array of claims tagged with veracity labels, accompanied by corresponding textual and visual evidence. Moreover, Factify2 [38] expanded the

**Table 1: Comparisons of fact-checking datasets. 'Claims' represents the number of claims.**

| Dataset | Multi-modal | Domain | Claims | Language | Evidence | Source | Misinformation |
|---|---|---|---|---|---|---|---|
| SciFact [43] | ✗ | Science | 1,409 | English | Text | Paper | - |
| PUBHEALTH [24] | ✗ | Health | 11,832 | English | Text | FACTWeb | - |
| FEVER [40] | ✗ | Multiple | 185,445 | English | Text | Wiki | - |
| FEVEROUS [4] | ✗ | Multiple | 87,026 | English | Text/Table | Wiki | - |
| CHEF [20] | ✗ | Multiple | 10,000 | Chinese | Text | Internet | - |
| FACTIFY [30] | ✓ | Multiple | 50,000 | English | Text/Image | FACTWeb | Collected |
| FACTIFY2 [38] | ✓ | Multiple | 50,000 | English | Text/Image | FACTWeb | Collected |
| NewsCLIPpings [1] | ✓ | Multiple | 85,360 | English | Text/Image | Internet | Synthetic |
| MOCHEG [46] | ✓ | Multiple | 21,184 | English | Text/Image | FACTWeb | Collected |
| MR2 [19] | ✓ | Multiple | 14,700 | Both | Text/Image | Internet | Collected |
| CMFC | ✓ | Multiple | 46,000 | Chinese | Text/Image | Multiple | Both |

paradigm by introducing a more comprehensive set of features and a more complex task framework. Hu *et al.* [19] collected relevant evidence on multi-modal fake news and constructed the MR2 dataset. MOCHEG [46] represented a large-scale fact-checking dataset, encompassing 21,184 claims, each assigned with corresponding evidence. However, it is noteworthy that existing datasets mostly focus on English and often narrowed down to a singular type of misinformation. Detailed information about the fact-checking detectors can be found in the *Supplementary Materials*.

**Knowledge Graph.** Knowledge Graph (KG) is a structured representation of information in the form of nodes and edges, where nodes represent entities or concepts, and edges represent the relationships between them [7]. Some studies [23, 15, 45] extract structured triples (head, relation, tail) from the post contents, and check them with the faithful triples in KG. However, existing fact-checking approaches that utilize knowledge graphs for aggregating entity knowledge and performing reasoning are limited to a single textual modality [17]. In our work, both textual and visual modalities of entity features from KG are taken into consideration.

## 3 THE CMFC DATASET

In contrast to previous work, we construct the first large-scale Chinese Multi-modal Fact-Checking (CMFC) dataset, subdivided into SCMF and CCMF. In these datasets, we assign each claim with one of two labels: pristine or falsified (similar to previous work [29, 1]). The construction of CMFC comprises three stages: claim data construction, evidence retrieval, and data preprocessing and analysis. During claim data construction, we select sources from which we extract statements and accompanying images. The process of evidence retrieval involves gathering relevant documents or sentences as evidence. Finally, we clean and analyze the dataset in the data preprocessing process.

### 3.1 Claim Data Construction

**CCMF:** We collect 10,000 fabricated claims that are naturally and widely disseminated from all active fact-checking websites in China. These claims encompass both text and image, along with authenticity labels. Typically, these claims originate from online speeches, public announcements, news articles, and social media platforms such as Weibo, WeChat, DouYin (TikTok), or various blogs. Authenticity labels are provided by fact-checkers. However, most claims

fact-checke d by fact-checkers are falsified, and solely relying on these claims would result in an imbalanced dataset. Therefore, we collect 16,000 real claims by scraping article titles or captions from four official news commentary websites. Detailed information about the source of claims can be found in the *Supplementary Materials*.

**SCMF:** While CCMF consists of real-world statements, the SCMF contains artificially created claims, generated by mutating sentences from real articles. In this type of threat, both the text and image of the claim originate from authentic news sources but are inaccurately matched, ensuring that unimodal text bias is not introduced into the dataset, which could potentially be captured by language models. We adopt a challenging, non-random image-text matching method to construct the dataset. We initially collect 10,000 pristine claims from real news websites. These statements include persons (*e.g.*, 'Musk', 'Berlusconi'), places (*e.g.*, 'Shanghai', 'Italy') or organizations (*e.g.*, 'International Court', 'Tesla Factory'), and more. By searching for this key information, we construct a corresponding falsified claim for each pristine claim by retrieving out-of-context misinformation images. It includes the following four types of fabrication: (1) By searching on the person entity, we retrieve news images related to that person but irrelevant to the original claim text; (2) By searching on the location entity, we retrieve news images related to the location mentioned in the original claim text but irrelevant to the text content; (3) By searching on the organization entity, we retrieve news images related to the organization mentioned in the original claim text but irrelevant to the text content; (4) An irrelevant image is randomly matched with the original claim text. By adopting these four fabrication methods, we built 10,000 falsified claims. Visualizations of the four methods, along with techniques to avoid retrieving images related to the text, are in the *Supplementary Materials*.

### 3.2 Evidence Retrieval

When verifying a claim, reporters need to find evidence that is relevant to the claim and help determine its authenticity label.

**CCMF:** Since the falsified claims in CCMF come from reliable fact-checking websites, for each article on these websites, we collect document text and document images corresponding to each claim. For pristine claims originating from real news websites, we crawl the corresponding news website documents. For a small number of documents that do not include images, we develop scripts to obtain

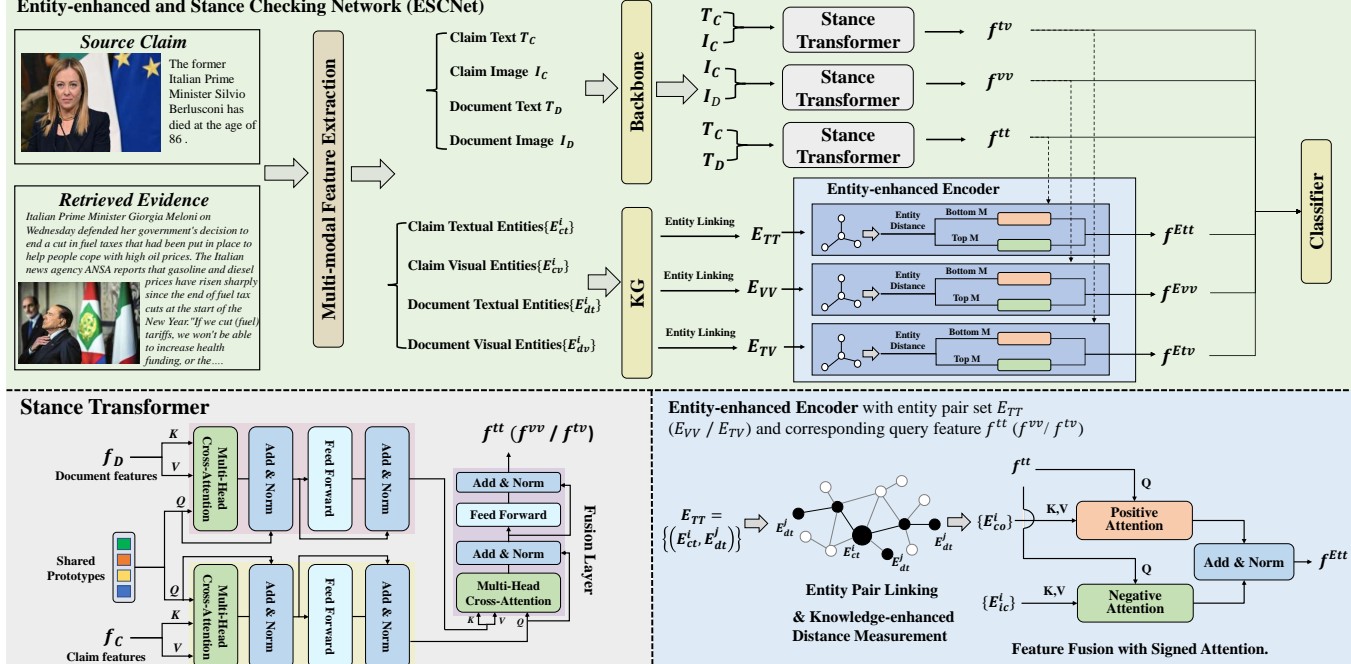

**Figure 4: The overall architecture of the proposed ESCNet.**

relevant images from the website by searching for documents and attaching them as corresponding document images.

**SCMF:** To detect the authenticity of the claims in SCMF, which include pristine and falsified ('out-of-context') images, we choose to manually extract evidence from web sources and use image-text pairs as queries to perform web searches. For text evidence, we use the Google Vision API to retrieve text evidence in reverse search mode using the claim image. The API returns the search result with the highest retrieval rank associated with the image, and we use it as text evidence. They may describe the content of the image and the news in which this image appears. For visual evidence, we use claim text as a text query to search for images. We use the Baidu Search API to perform image searches, where the search results are not always an exact match for the text query. Even if it is not entirely related to the event, it can serve as a useful clue about the type of image that might be associated with the topic.

### 3.3 Data Preprocessing and Analysis

The initial CMFC dataset has many crawling issues, such as being unable to retrieve articles, or the content not being text. We remove such instances. Next, we check the dataset for duplicates and all duplicates are included in the training split of the dataset. We clean the content of claim texts, removing keywords that could leak information (such as 'debunk', 'truth', *etc*). Besides, Chinese fact-checkers tend to express non-factual claims in rhetorical questions. To avoid the impact of tone and symbols, we convert these into declarative statements. Finally, we built the Chinese Multimodal Fact Checking dataset (CMFC) containing 46,000 claims, of which SCMF accounted for 20,000 and CCMF contained 26,000. As

shown in Figure 2, More than 30% of the claims in both datasets belong to the health domain, as many of the Chinese-language fact-checking articles focus on refuting falsified information related to 'COVID-19'. The political domain account for the second, reflecting the continued influence and attention of political topics in society. Figure 3 proves the above observation. Moreover, we compare the CMFC with other datasets in Table 1: 'Multi-modal' represents whether the dataset is multi-modal; 'Domain' represents the fields involved in the dataset; 'Evidence' means the type of evidence used, which can be text (includes metadata), table or image. 'Source' means where the evidence is collected from, such as Wikipedia (Wiki), fact-checking websites (FACTWeb) and the Internet. 'Misinformation' represents the source of misinformation within the multi-modal dataset. Compared to other datasets, we can attribute the strength of CMFC to several aspects: (1) The first large-scale Chinese multi-modal fact-checking dataset; (2) Covering two types of misinformation in the multi-modal fact-checking task; (3) The dataset is of substantial size, with claims and their corresponding evidence document sourced from a diverse range of platforms and domains. More analysis and examples can be found in the *Supplementary Materials*.

## 4 METHOD

### 4.1 Model Overview

As illustrated in Figure 4, our ESCNet mainly consists of three parts: Multi-modal Feature Extraction Module, Stance Transformer and Entity-enhanced Encoder. The details about the three parts are described in the following subsections.

## 4.2 Multi-modal Feature Extraction Module

Given a source claim that includes images and text, we extract the following four features: (1) Claim Text $T_C$, i.e., the text portion of the source claim. (2) Claim Textual Entities $\{E_{ct}^i\}$, typically containing named entities such as person and places. These entities are crucial for understanding news semantics and aiding in fact-checking. (3) Claim Visual Entities $\{E_{cv}^i\}$, which include named entities in the claim images. Similar to text, news images also contain visual entities that are vital for semantic understanding and the detection of falsified claims. (4) Claim Image $I_C$, which includes the visual CNN features in the claim images. We utilize ResNet-50, without the final layer, to encode the images as initial feature maps, which are then transformed into visual feature sequences through convolutional blocks and reshaping operations. Similarly, we also extract the following features with the retrieved document evidence: Document Text $T_D$; Document Textual Entities $\{E_{dt}^i\}$; Document Visual Entities $\{E_{dv}^i\}$; Document Image $I_D$.

## 4.3 Stance Transformer

After extracting multi-modal cues, we employ three Stance Transformers to simulate human hierarchical reasoning process and model the high-order unimodal and cross-modal correlation, including text reasoning ($T_C$ and $T_D$), image reasoning ($I_C$ and $I_D$), and cross-modal reasoning ($T_C$ and $I_C$). Taking text reasoning as an example: the Stance Transformer first introduces a set of Shared Prototypes as queries to guide the reconstruction of the feature representations of the claim and its corresponding evidence. The advantages are twofold: (1) Regardless of the length of the input evidence document, the final output is the length of the Shared Prototypes, which significantly reduces computational complexity and prevents information loss due to excessively long evidence. (2) By projecting the two features into the same vector space, we promote a better comparison and fusion of features. Specifically, a set of Shared Prototypes $\{P_i\}$ are composed of $k$ vectors of length $L$, randomly initialized. A shared multi-head attention transformer layer (*MultiHead_Shared*) [42] is utilized to reconstruct the representations. We take each prototype as a query, while the claim or document features simultaneously as keys and values:

$$\hat{T}_C = MultiHead\_Shared(\{P_i\}, T_C, T_C)$$
$$\hat{T}_D = MultiHead\_Shared(\{P_i\}, T_D, T_D) \tag{1}$$

Then, we fuse the reconstructed claim feature representation $\hat{T}_C$ with the evidence representation $\hat{T}_D$ through the fusion layer, to obtain the stance representation of the evidence $T_D$ towards the claim $T_C$ in the text reasoning.

$$f^{tt} = MultiHead\_Fusion(\hat{T}_C, \hat{T}_D, \hat{T}_D) \tag{2}$$

where *MultiHead_Fusion* is a multi-head attention transformer layer. Following the above operations, we obtain the stance representations $f^{vv}$ in image branch and $f^{tv}$ in cross-modal branch.

## 4.4 Entity-enhanced Encoder

Multi-modal entity correlation is a key indicator of claims. Entity-enhanced Encoder (EeE) links entities to acquire higher-order semantic information from KG, conducting multi-modal fact-checking at the knowledge level. Firstly, it identifies visual and textual entities from images and texts originating from source claims and evidence documents, followed by linking to generate a cross-modal entity pair set $E_{TV}$, and two unimodal entities pair sets ($E_{TT}$ and $E_{VV}$). A new knowledge-related reasoning strategy is proposed to measure the knowledge-enhanced distance of each entity pair sets and build knowledge-enhanced entity representations. Next, it applies negative or positive signed attention mechanisms, using the stance reasoning features from the Stance Transformer to select entity pairs with semantic inconsistencies and consistencies.

**Entity Pair Linking.** By jointly considering the pairwise relationships within and across the entity sets of claim and document, the intra-modal and cross-modal entity correlations are explored. The EeE links all possible pairings into two intra-modal entity pair sets $E_{TT}$ and $E_{VV}$, respectively as follows:

$$E_{TT} = (E_{ct}^i, E_{dt}^j) : 1 \le i \le N_{ct}, 1 \le j \le N_{dt}$$
$$E_{VV} = (E_{cv}^i, E_{dv}^j) : 1 \le i \le N_{cv}, 1 \le j \le N_{dv} \tag{3}$$

where $N_{ct}$, $N_{dt}$, $N_{cv}$, and $N_{dv}$ represent the number of entities in the corresponding entity sets. Similarly, EeE constructs the cross-modal entity pair set $E_{TV}$ as follows:

$$E_{TV} = (E_{ct}^i, E_{cv}^j) : 1 \le i \le N_{ct}, 1 \le j \le N_{cv} \tag{4}$$

It then uses the following knowledge-enhanced reasoning strategy to measure the knowledge-enhanced distance of each pair sets.

**Knowledge-enhanced Distance Measurement.** Given an entity pair $(E^u, E^v)$ from arbitary set of $E_{TT}$, $E_{VV}$ or $E_{TV}$, we propose a novel metric $D(E^u, E^v)$ to measure the knowledge-enhanced distance of the two entities on a pretrained knowledge gragh. Different from the metrics in previous works that only considered pairwise feature distance without background contextual knowledge in the knowledge gragh, the metric $D$ is capable to leverage the feature distance in the embedding space as well as the graph distance on the KG topology, which is more appropriate to model the semantic relevant. EeE firstly finds a shortest semantic relevant path $\pi$ in the KG which connects $E^u$ and $E^v$:

$$\pi : E^u = E^{w_0} \to E^{w_1} \to \ldots \to E^{w_n} = E^v \tag{5}$$

Where $n$ denotes the number of the entities in $\pi$. EeE realizes real-time semantic relevant path searching in a large KG with a modified version of Floyd-Warshall algorithm [18] to trade off runtime efficiency, memory consumption and path optimality. *More details of the proposed algorithm can be found in Supplementary Materials.* After obtaining the optimal path $\pi$, the module refines the knowledge-enhanced entity representation $h^u$ for entity $E^u$:

$$h^u = \frac{1-\alpha}{1-\alpha^{n+1}} \sum_{i=0}^{n} \alpha^i g^{w_i} \tag{6}$$

where $g^{w_i}$ is the feature embedding of entity $E^{w_i}$ in the path. $\alpha$ denotes a weight coefficient ($\alpha = 0.9$). It's worth noting that $h^u$ is a path-aware representation, i.e., a different pair $(E^u, E^{v'})$ with path $\pi'$ will yield a different value of $h^u$. Intuitively, $h^u$ absorbs complementary contextual knowledge from all entities along the path $\pi$ by weighted averaging their KG embeddings, with the weight exponentially descending to dilute their contributions as the graph

distance increases. Symmetrically, EeE refines the representation $h^v$ for entity $E^v$ as following:

$$h^v = \frac{1-\alpha}{1-\alpha^{n+1}} \sum_{i=0}^{n} \alpha^{n-i} g^{w_i} \qquad (7)$$

The knowledge-enhanced distance $D(E^u, E^v)$ is calculated as the Euclidean distance between $h^u$ and $h^v$

$$D(E^u, E^v) = \|h^u - h^v\|_2 \qquad (8)$$

The concatenated feature $[h^u; h^v]$ is treated as the semantic relevant entity representation for pair $(E^u, E^v)$. The semantic relevant entity representation and knowledge-enhanced distance are further utilized in exploring feature fusion.

**Feature Fusion with Signed Attention.** The EeE models the high-order knowledge-enhanced entity relevance in each entity pair set by filtering the bottom/top $m$ distance pairs as the most consistent/inconsistent pairs and applying positive/negative signed attention. Signed attention allows the model to not only consider positive correlations between elements (*e.g.*, queries and keys), but also to recognize opposing or contrasting semantics, which can be beneficial for the fact-checking task [37]. Taking the entity pair $E_{TT}$ in the text branch as an example, we introduce our enhanced method: First, we use a knowledge graph to encode each entity and measure their knowledge-enhanced distance for each pair of entity representations in $E_{TT}$. We retain the $m$ pairs with the smallest distance as the consistency entity pair subset $\{E_{co}^i\}$ and their corresponding distance values $\{D_{pos}^i\}$, and the $m$ pairs with the largest distance as the inconsistency entity pair subset $\{E_{ic}^i\}$ and their corresponding distance values $\{D_{neg}^i\}$. Then, we further fuse the selected knowledge-enhanced entity representations with stance reasoning features by utilizing the signed attention mechanism, so as to simultaneously capture high-order consistent and inconsistent entity relevance. Specifically, the EeE first adopts positive attention to capture consistent relevance, relative to the latter content. It takes the text reasoning feature $f^{tt}$ from the Stance Transformer as the query to calculate the consistent relevance as follows:

$$\alpha_{pos}^i = \text{Softmax}\left(f^{tt}\{E_{co}^i\}^T / \sqrt{d_e}\right)$$
$$f_{pos}^{Ett} = \left(\sum_{i=1}^{k} \frac{\alpha_{pos}^i}{D_{pos}^i}\{E_{co}^i\}\right) \bigg/ \left(\sum_{i=1}^{k} \frac{\alpha_{pos}^i}{D_{pos}^i}\right) \qquad (9)$$

where $d_e$ is the dimension of $\{E_{co}^i\}$. $\alpha_{pos}^i$ denotes the positive attention coefficients. A larger $\alpha_{pos}^i$ indicates that the entity pair is more positively semantically associated with the post content. Note that we re-weight the coefficients with $\{D_{pos}^i\}$ to incorporate semantic relevant distances into the consistency representation. The entity pairs with shorter semantic relevant distances have a greater impact on the learning of consistency relevance.

Simultaneously, the EeE utilizes negative attention to estimate the inconsistency representation $f_{neg}^{Ett}$.

$$\alpha_{neg}^i = -\text{Softmax}\left(-f^{tt}\{E_{ic}^i\}^T / \sqrt{d_e}\right)$$
$$f_{neg}^{Ett} = \left(\sum_{i=1}^{k} \alpha_{neg} D_{neg}^i \{E_{ic}^i\}\right) \bigg/ \left(\sum_{i=1}^{k} \alpha_{neg} D_{neg}^i\right) \qquad (10)$$

**Table 2: Performance comparison to the state-of-the-art methods on CCMF, SCMF and NewsCLIPpings datasets.**

| | Methods | Acc | Prec | Rec | F1 |
|---|---|---|---|---|---|
| CCMF | UofA-Truth | 0.745 | 0.749 | 0.761 | 0.755 |
| | Logically | 0.737 | 0.724 | 0.720 | 0.722 |
| | CCN | 0.793 | 0.853 | 0.738 | 0.791 |
| | MAFN | 0.813 | 0.851 | 0.767 | 0.807 |
| | INO | 0.826 | 0.842 | 0.791 | 0.816 |
| | END | 0.834 | 0.825 | 0.835 | 0.830 |
| | Triple-Check | 0.832 | 0.823 | 0.829 | 0.826 |
| | **ESCNet** | **0.862** | **0.857** | **0.852** | **0.854** |
| SCMF | UofA-Truth | 0.702 | 0.711 | 0.702 | 0.706 |
| | Logically | 0.717 | 0.732 | 0.717 | 0.724 |
| | CCN | 0.767 | 0.770 | 0.767 | 0.769 |
| | MAFN | 0.782 | 0.773 | 0.783 | 0.778 |
| | INO | 0.804 | 0.811 | 0.804 | 0.807 |
| | END | 0.810 | 0.826 | 0.810 | 0.818 |
| | Triple-Check | 0.813 | 0.829 | 0.812 | 0.820 |
| | **ESCNet** | **0.849** | **0.840** | **0.844** | **0.842** |
| NewsCLIPpings | UofA-Truth | 0.768 | 0.768 | 0.768 | 0.768 |
| | Logically | 0.786 | 0.791 | 0.786 | 0.788 |
| | CCN | 0.847 | 0.853 | 0.852 | 0.852 |
| | MAFN | 0.802 | 0.813 | 0.802 | 0.808 |
| | INO | 0.823 | 0.834 | 0.823 | 0.828 |
| | END | 0.833 | 0.838 | 0.833 | 0.835 |
| | Triple-Check | 0.848 | 0.850 | 0.851 | 0.851 |
| | **ESCNet** | **0.879** | **0.872** | **0.875** | **0.874** |

We re-weight the coefficients with $\{D_{neg}^i\}$ to incorporate relevant distances into the inconsistency representation, where entity pairs with larger relevant distances have a greater impact on the learning of inconsistency relevance. Finally, the representations $f_{neg}^{Ett}$ and $f_{pos}^{Ett}$ are concatenated to form the knowledge-enhanced entity pair reasoning feature $f^{Ett}$ of the entity pair set $E_{TT}$. Similarly, EeE obtains the relevant representations $f^{Evv}$, $f^{Etv}$ of $E_{VV}$ and $E_{TV}$ with the same mechanism. The extracted features $f^{tt}$, $f^{vv}$, $f^{tv}$, $f^{Ett}$, $f^{Evv}$ and $f^{Etv}$ are finally contacted and fed into the Classifier for fact-checking. The details of the Classifier are provided in the *Supplementary Materials*.

## 5 EXPERIMENTS

### 5.1 Experimental Settings

**Dataset.** We evaluate the proposed method ESCNet on our two datasets which contain both pristine and falsified claims. In order to evaluate our ESCNet more comprehensively, we introduce a large-scale English fact-checking dataset, NewsCLIPpings. We divide these datasets into training, validation and testing sets according to 6:2:2 and apply the accuracy score, precision, recall and F1 score as our evaluation metric, which is widely used for binary classification tasks. The statistical details of NewsCLIPpings are reported in *Supplementary Materials*.

**Implementation Details.** Regarding image content, we employ the Baidu platform APIs to recognize and extract these entities, which are treated as visual entity mentions and are linked to the

corresponding entities in the Knowledge Graph (KG). Regarding text content, the named entity linking tools bert-base-chinese-ner [35] and Tagme are applied to link the ambiguous entity mentions in the texts. Freebase [5] is introduced as the background KG, where the pre-trained embeddings of the entities with 50 dimensions are provided by the method [6]. Moreover, in the textual backbone, we set the length of the input text to at most 512 tokens, and utilize the pre-trained Chinese BERT model [11] to initialize the word embeddings with 768 dimensions. We use the pre-trained ResNet-50 model as visual backbone. In terms of parameter setting, we set the learning rate of the overall framework to $2e^{-4}$. The batch size of the input is 64. The value of $m$ is selected from {1, 2, 3, 4, 5}.

## 5.2 Comparison to State-of-the-Art Approaches

In order to evaluate the ESCNet, we compare it with the following state-of-the-art methods on different fact-checking datasets, including END [46], CCN [1], UofA-Truth [13], MAFN [39], INO [48], Triple-Check [14] and Logically [16]. In Table 2, we can observe that the proposed ESCNet achieves the best performance of 86.2%, 84.9% and 87.9% accuracy respectively on three datasets. The results demonstrate the effectiveness of the proposed method. Among these compared methods, Logically and INO obtain relatively high performance on these datasets, demonstrating the powerful ability to capture consistency between texts and images with pre-trained CLIP [33]. END pairs each piece of evidence with the input claim and detects the stance of the evidence towards the claim. CCN achieves a relatively high score by adopting a checking architecture, demonstrating the importance of introducing external complementary knowledge information (such as entities). MAFN experiments with a inter-modality and intra-modality fusion of textual and visual embeddings. UofA-Truth breaks the task into text and image entailment sub-tasks, using sentence BERT for text embeddings and Xception for image embeddings. These models achieve good performance by using BERT as the backbone. Triple-Check proposes a model that employs pre-trained DeBERTa for text embeddings and Swinv2 for image embeddings, fused through a co-attention block. It outperforms most of the compared methods, demonstrating the effectiveness of enhancing the features with the attention network. Compared to the aforementioned methods, we can attribute the strength of ESCNet to several aspects: 1) The usage of the BERT model as a part of the backbone networks, results in a strong textual representation. 2) We extract six types of reasoning information from source claims and retrieved evidence, which are more suitable for fact-checking. 3) The Stance Transformer module can effectively detect the stance of the evidence towards the claim for different types of evidence modalities. 4) The EeE links entities from the texts and images to the KG to acquire high-level entity information and conduct fact-checking at the knowledge level.

## 5.3 Ablation Studies

**Analysis of detailed features.** The results in Table 3 show the influence of different detailed features. The six columns from top to bottom correspond to without the six features in Figure 4: the stance semantic reasoning features $f^{tt}$, $f^{vv}$, $f^{tv}$ and the knowledge-enhanced entity pair features $f^{Ett}$, $f^{Evv}$ and $f^{Etv}$. We demonstrate

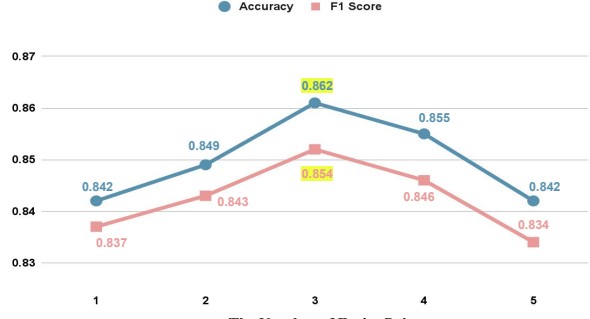

**Figure 5: Evaluation of the number of entity pairs on CCMF.**

**Table 3: Evaluation of the influence of different features of ESCNet on CCMF dataset.**

| Methods | Acc | Prec | Rec | F1 |
|---|---|---|---|---|
| w/o f_tt | 0.744 | 0.749 | 0.762 | 0.755 |
| w/o f_vv | 0.801 | 0.799 | 0.815 | 0.806 |
| w/o f_tv | 0.817 | 0.813 | 0.797 | 0.805 |
| w/o f_Ett | 0.832 | 0.825 | 0.840 | 0.832 |
| w/o f_Evv | 0.844 | 0.838 | 0.830 | 0.834 |
| w/o f_Etv | 0.853 | 0.845 | 0.846 | 0.845 |
| **ESCNet** | **0.862** | **0.857** | **0.852** | **0.854** |

**Table 4: Evaluation of the influence of different components of Stance Transformer on the CCMF dataset.**

| Methods | Acc | Prec | Rec | F1 |
|---|---|---|---|---|
| w/o Stance | 0.819 | 0.836 | 0.782 | 0.808 |
| w/o Shared-P | 0.834 | 0.838 | 0.808 | 0.823 |
| w/o Fusion | 0.854 | 0.850 | 0.840 | 0.845 |
| **ESCNet** | **0.862** | **0.857** | **0.852** | **0.854** |

**Table 5: Evaluation of the effectiveness of Entity-enhanced Encoder on the SCMF dataset.**

| Methods | Acc | Prec | Rec | F1 |
|---|---|---|---|---|
| w/o Entity Pos | 0.809 | 0.834 | 0.768 | 0.799 |
| w/o Entity Neg | 0.823 | 0.819 | 0.836 | 0.828 |
| w/o Enhanced Path | 0.830 | 0.827 | 0.811 | 0.819 |
| **ESCNet** | **0.849** | **0.840** | **0.844** | **0.842** |

the benefits of each decision feature, which highlights the importance of integrating all modalities for multi-modal fact-checking. Removing text stance reasoning features $f^{tt}$ or image stance reasoning features $f^{vv}$ significantly reduces performance, which demonstrates the importance of these two features. The impact of removing entities is relatively smaller and might be because some redundant information is present in the evidence document text, or sometimes generic named entities do not contribute to the checking of the claim statements.

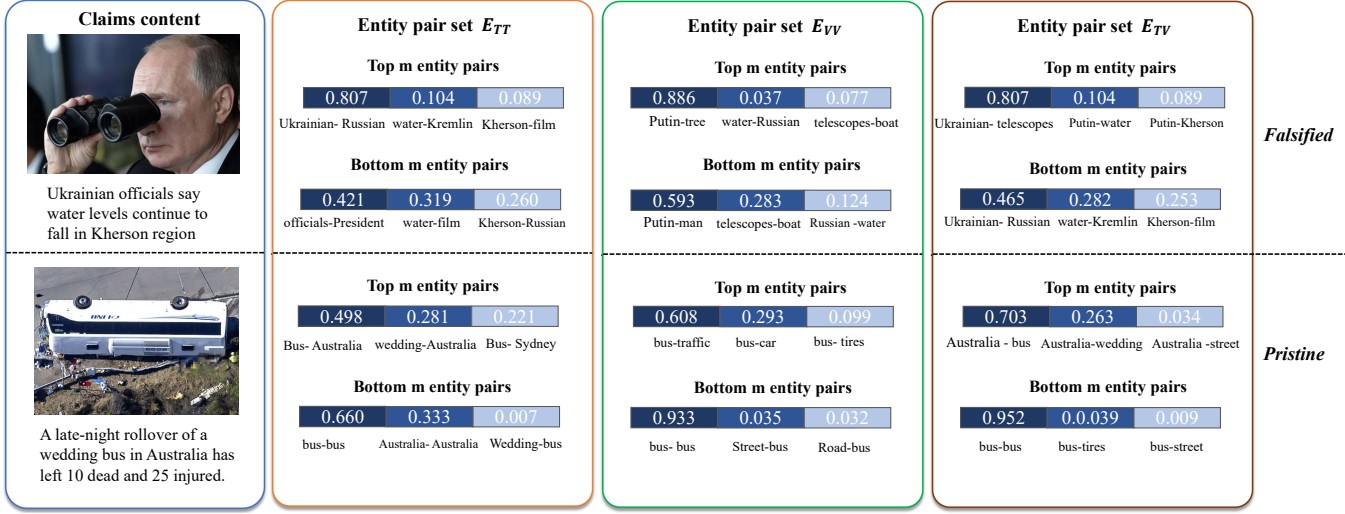

**Figure 6: Visualization of entity pair attention distribution in Entity-enhanced Encoder.**

**Analysis of Stance Transformer.** We conduct experiments to analyze the effectiveness of the proposed Stance Transformer in Table 4. *w/o Stance* denotes ESCNet without using the Stance Transformer for feature fusion (replacing with concatenate operation), *w/o Shared-P* refers to Stance Transformer without using Shared Prototypes to reconstruct claim and document representations (replacing with a shared FC layer). *w/o Fusion* denotes Stance Transformer without using fusion layer (replacing with concatenate operation). We observe that the Stance Transformer contributes to performance improvement. The comparative results highlight the advantages of projecting both features into the same vector space, which facilitates a comprehensive feature fusion.

**Analysis of Entity-enhanced Encoder.** The experimental results in Table 5 show that the Entity-enhanced Encoder improves performance. *w/o Entity Pos* represents Entity-enhanced Encoder without using the subset and features of consistent entity pairs, while *w/o Entity Neg* represents EeE without using the subset and features of inconsistent entity pairs. The results suggest that the signed attention network can effectively capture and fuse the consistency and inconsistency of knowledge-enhanced entity pairs, each of which holds significant influence for multi-modal fact-checking. *w/o Enhanced Path* denotes EeE without using the knowledge-enhanced distance (replacing with direct Euclidean distance), demonstrating the benefits of knowledge-enhanced distance measurement. We also tried different values of Top $m$, *i.e.*, the number of the top/bottom distance entity pairs. A small $m$ increases the risk of discarding entity pairs' information, while a large $m$ increases the risk of introducing irrelevant noise. As shown in Figure 5, $m = 3$ leads to the best performance.

### 5.4 Qualitative Evaluation

Figure 6 (The corresponding evidence documents can be found in Figure 8) shows $m$ ($m = 3$) pairs of entities with the top (bottom) knowledge-enhanced distance and the corresponding distribution of negative attention scores (distribution of positive attention

scores) across varying entity pair sets. We can find that: (1) For falsified claims, the differences between entity pairs are large regardless of whether they are the entity with the largest distance or the entity with the smallest distance, and the distribution of negative attention scores corresponding to top $m$ entity pairs is not balanced and tends to be concentrated in the most inconsistent entity pairs (e.g., 'Putin-tree'), whereas the distribution of positive attention scores corresponding to bottom $m$ entity pairs is relatively balanced. (2) For pristine claims, whether it is the entity with the largest distance or the entity with the smallest distance, the difference between entity pairs is relatively small, and even duplicate entity pairs (e.g., 'bus-bus') are often found. Top $m$ entity pairs correspond to relatively balanced negative attention scores, while bottom $m$ entity pairs correspond to an unbalanced distribution of positive attention scores, which tend to be concentrated in the most consistent entity pairs (e.g., 'bus-bus'). This suggests that the Entity-enhanced Encoder can effectively capture and fuse the consistency and inconsistency of knowledge-enhanced entity pairs, each of which holds significant influence for multi-modal fact-checking. Additional qualitative experiments and discussions can be found in the *Supplementary Materials*.

## 6 CONCLUSION

In this work, we construct the first large-scale, multi-domain Chinese Multi-modal Fact-Checking (CMFC) dataset. The CMFC covers all types of misinformation and is divided into two sub-datasets, CCMF and SCMF. To establish a baseline performance, we introduce a novel Entity-enhanced and Stance Checking Network (ESCNet), which includes Multi-modal Feature Extraction Module, Stance Transformer, and Entity-enhanced Encoder (EeE). The proposed ESCNet jointly model both stance semantic reasoning features and knowledge-enhanced entity pair features, facilitating the learning of effective semantic-level and knowledge-level claim representations. Extensive experiments on Chinese and English fact-checking datasets demonstrate the effectiveness of the proposed method.

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

## A CLASSIFICATION

ESCNet leverages stance semantic reasoning features and knowledge-enhanced entity pair features to jointly perform multi-modal fact-checking. We perform average pooling on all features along the sequence dimension. The stance semantic reasoning features $f^{tt}$, $f^{vv}$, $f^{tv}$ and the knowledge-enhanced entity pair features $f^{Ett}$, $f^{Evv}$ and $f^{Etv}$ are finally concatenated to form the discriminative claim representation, which is further transformed by an FC layer with *Softmax* activation function to predict results as following:

$$\hat{y} = \sigma(W_c^T [f^{tt}; f^{vv}; f^{tv}; f^{Ett}; f^{Evv}; f^{Etv}] + b_c) \tag{11}$$

where $W_c$ and $b_c$ are the parameters of the classifier layer. We then use the cross-entropy loss function as the loss for the whole model, which is formulated as described below:

$$L = -\sum_{i=1}^{|M|} y_i \log(\hat{y}_i) \tag{12}$$

where $M$ refers to the number of distinct label categories.

## B KNOWLEDGE-ENHANCED DISTANCE

In this section, we elaborate the detail of shortest path finding algorithm mentioned in Section 4.4. Given an unweighted undigraph $G = (V, E)$ and a pair of vertices $v_s, v_t \in V$ as query, it yields a sub-optimally shortest path $\pi : v_s \to \dots \to v_t$ with time complexity $\Theta(|\pi|)$ and relatively low memory consumption. Considering the KG we adopt is quite large (with over 3 million vertices), the algorithm is composed of two steps. In the first step, we perform a hierarchical Floyd-Warshall algorithm offline to extract and store essential path reconstruction metadata. In the second step, we use the extracted metadata to find the path for each vertex pair efficiently online. This section will first briefly review the Floyd-Warshall algorithm, and further introduce the metadata extraction strategy and the pair-wise path finding strategy in our algorithm.

**The Floyd-Warshall Algorithm.** Given an unweighted graph $G = (V, E)$ with $n$ vertices $V = \{v_i\}_{i=1}^n$, the Floyd-Warshall algorithm [12] computes the pair-wise shortest distance matrix $D \in \mathbb{R}^{n \times n}$ and the path reconstruction matrix $C \in \mathbb{R}^{n \times n}$ with time complexity $\Theta(n^3)$ as illustrated in Algorithm 1. Each element $D(i, j)$ of $D$ stores the shortest distance between vertex pair $(v_i, v_j)$. The matrix $C$ contains information for path reconstruction, with which one can reconstruct the actual path between two connected vertices, as illustrated in Algorithm 2. In general, we can pre-compute the reconstruction matrix $C$ for a graph $G$ offline, and adopt it for efficient online path finding with linear time complexity and $\Theta(n^2)$ space complexity.

**Offline Path Reconstruction Metadata Extraction.** Since the adopted KG is in large scale, it's impractical to perform the standard Floyd-Warshall algorithm on the whole graph. We instead partition the graph $G$ into several smaller sub-graphs, and record the path reconstruction metadata within and among sub-graphs for better runtime and memory efficiency, which are further adopted for online path finding.

Specifically, the vertex set $V$ of graph $G$ is partitioned into $M = 94406$ disjoint groups $\{V_i\}_{i=1}^M$, such that (1) the size of each group $|V_i| \leq 512$, and (2) all vertices within a group $V_i$ is pair-wisely connected. The partition process is conducted by first sorting

---

**Algorithm 1:** The Floyd-Warshall Algorithm

**Input:** $G = (V, E)$.
**Output:** $D \in \mathbb{R}^{n \times n}$ and $C \in \mathbb{R}^{n \times n}$.
    ▷ Initialize $D$ and $C$ Fill $D$ with $\infty$; Fill $C$ with 0;
**for** $(v_i, v_j) \in E$ **do**
    $D(i, j) \leftarrow 1; C(i, j) \leftarrow j$;
**end**
**for** $v_i \in C$ **do**
    $D(i, i) \leftarrow 0; C(i, i) \leftarrow i$;
**end**
    ▷ Standard Floyd-Warshall Algorithm **for** $k \leftarrow 1 \dots n$ **do**
    **for** $i \leftarrow 1 \dots n$ **do**
        **for** $j \leftarrow 1 \dots n$ **do**
            **if** $D(i, j) > D(i, k) + D(k, j)$ **then**
                $D(i, j) \leftarrow D(i, k) + D(k, j)$;
                $C(i, j) \leftarrow C(i, k)$;
            **end**
        **end**
    **end**
**end**
**return** $D$ and $C$;

---

all vertices by their degrees in ascending order and then greedily merging connected vertex pairs to form the groups. We define two groups $V_i, V_j$ are *adjacent* if there exists an edge $(u, v) \in E$ such that $u \in V_i, v \in V_j$, or *connected* if there exists a path $u \to \dots \to v$ such that $u \in V_i, v \in V_j$. Moreover, we categorize the groups into two types based on their scale, *i.e.*, the *big group* with size $\geq 100$, and the *small group* with size $< 100$, satisfying that (1) every two *big groups* are connected and (2) each *small group* is adjacent to at least one *big group*. The total number of *big groups* is $M_b = 2364$.

With this partition, we then compute the path reconstruction metadata for the whole KG. Particularly, we first perform the Floyd-Warshall algorithm for each group $V_i$ to obtain corresponding reconstruction matrix $C_i$. Furthermore, we create a hyper-graph $\tilde{G}$ with all *big groups* $\{V_i\}_{i=1}^{M_b}$ as its hyper-vertices, and perform Floyd-Warshall algorithm on the hyper-graph $\tilde{G}$ to obtain its reconstruction matrix $\tilde{C} \in \mathbb{R}^{M_b \times M_b}$. Lastly, we store (1) the adjacent matrix of hyper-graph $\tilde{G}$, (2) the edges between each *small group* and all its adjacent *big groups* and (3) the reconstruction matrices $\{C_i\}$ and $\tilde{C}$ as the path reconstruction metadata of our adopted KG, which overall occupies 1.3GB memory.

**Online Path Finding.** With the extracted path reconstruction metadata, we are able to find the shortest path $\pi$ for a vertex pair $v_s, v_t \in V$ with efficiency online. The process is simply a path traversal in each group and in the hyper-graph under the guidance of the reconstruction matrices $\{C_i\}$ and $\tilde{C}$, which can be demonstrated by the pseudo code in Algorithm 3. Since each vertex in $\pi$ is visited exactly once, the path finding achieves linear time complexity of $\Theta(|\pi|)$. The obtained path is not guaranteed to be optimal, but is reasonable enough for calculating knowledge-enhanced distance. To avoid potentially extra long path in practice, we perform path finding simultaneously from both $v_s$ and $v_t$, and prematurely stop the process if current path contains more than 40 vertices. Such

**Algorithm 2:** Path Reconstruction with $C$

**Input:** Vertex pair query $v_i, v_j \in V$.
**Output:** The shortest path $\pi \subset V$ connecting $v_i, v_j$.
**if** $C(i, j) = 0$ **then** // not connected
    **return** empty path;
**end**
$\pi \leftarrow [v_i]$;
**while** $v_i \neq v_j$ **do**
    $i \leftarrow C(i, j)$;
    Append $v_i$ to $\pi$;
**end**
**return** $\pi$;

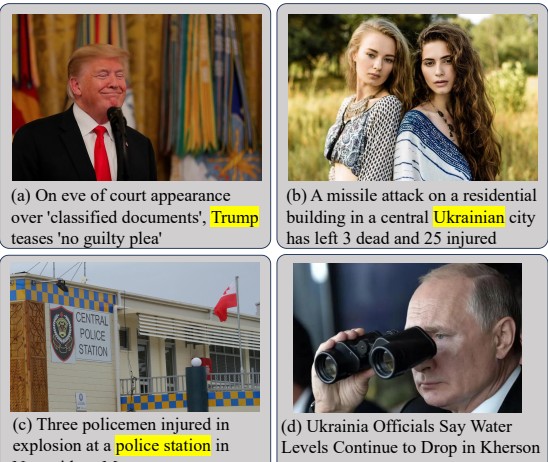

(a) On eve of court appearance over 'classified documents', Trump teases 'no guilty plea'

(b) A missile attack on a residential building in a central Ukrainian city has left 3 dead and 25 injured

(c) Three policemen injured in explosion at a police station in Naypyidaw, Myanmar

(d) Ukrainia Officials Say Water Levels Continue to Drop in Kherson Region

**Figure 7: Four types of falsified information from SCMF (Chinese is translated into English). Entities used to perform a search for images are highlighted in yellow.**

strategy does not affect the calculation of knowledge-enhanced distance since vertices with overly large graph distance have negligible contribution due to the exponential decay of their weights. In rare cases where $v_s$ and $v_t$ are not connected, we simply aggregate adjacent vertices in respective neighbor for semantic relevance measurement. We implement the whole algorithm with the Cython[1] language for lower latency and better parallelism.

## C MORE ANALYSIS ABOUT THE CMFC

### C.1 Data Source

As shown in Table 6, we cover all active fact-checking websites in China. These websites include 'Piyao', 'Mingcha', 'Youjv', and others, which have verified falsified claims across various platforms. They cover different domains and provide compelling documents and verdicts. Furthermore, to establish a comprehensive CCMF

---

[1] https://cython.org/

**Algorithm 3:** Path Finding in the KG

**Input:** Vertex pair query $v_s, v_t \in V$.
**Output:** The shortest path $\pi \subset V$ connecting $v_s, v_t$.
**if** either of $v_s, v_t$ is isolated vertex **then**
    **return** empty path;
**end**
$\pi \leftarrow$ empty path;
$V_{i_s}, V_{i_t} \leftarrow$ the groups that contain $v_s, v_t$;
**if** $i_s = i_t$ **then** // in the same group
    Reconstruct $\pi : v_s \rightarrow v_t$ with $C_{i_s}$;
    **return** $\pi$;
**end**
▷ Handle edge cases where either $V_{i_s}$ or $V_{i_t}$ is *small group* **if** $V_{i_s}$ is a *small group* **then**
    Randomly choose an edge $(u, w)$ that connects $V_{i_s}$ and a *large group* $V_k$;
    Reconstruct $\pi' : v_s \rightarrow u$ with $C_{i_s}$;
    Append $\pi'$ to $\pi$;
    $v_s \leftarrow w$; $i_s \leftarrow k$;
**end**
$\pi_r \leftarrow$ empty path; // potential residual path
**if** $V_{i_t}$ is a *small group* **then**
    Randomly choose an edge $(u, w)$ that connects a *large group* $V_k$ and $V_{i_t}$;
    Reconstruct $\pi_r : w \rightarrow v_t$ with $C_{i_t}$;
    $v_t \leftarrow u$; $i_t \leftarrow k$;
**end**
    ▷ Now $V_{i_s}$ and $V_{i_t}$ are both *large groups* **while** $i_s \neq i_t$ **do**
    $k \leftarrow \tilde{C}(i_s, i_t)$; // the next hyper-vertex
    Randomly choose an edge $(u, w)$ that connects $V_{i_s}$ and $V_k$;
    Reconstruct $\pi' : v_s \rightarrow u$ with $C_{i_s}$;
    Append $\pi'$ to $\pi$;
    $v_s \leftarrow w$; $i_s \leftarrow k$;
**end**
Append $\pi_r$ to $\pi$;
**return** $\pi$;

**Table 6: Statistics of data sources.**

| Website | Domain | URL | Label |
|---|---|---|---|
| Mingcha | Multiple | www.factpaper.cn | falsified |
| Youjv | Politics | chinafactcheck.com | falsified |
| Piyao | Multiple | www.piyao.org.cn | falsified |
| Kexue | Science | piyao.kepuchina.cn | falsified |
| Shanghai | Multiple | piyao.jfdaily.com | falsified |
| Jiaozhen | Health | vp.fact.qq.com | falsified |
| Pengpai | Multiple | www.thepaper.cn | pristine |
| Chinanews | Multiple | m.chinanews.com | pristine |
| Xinhua | Multiple | www.news.cn | pristine |
| Huanqiu | Multiple | www.huanqiu.com | pristine |

| Claim Text | Claim Image | Text Evidence | Image Evidence | Truthfulness |
|---|---|---|---|---|
| **#1:** Ukrainia Officials Say Water Levels Continue to Drop in Kherson Region |  | Russian President Vladimir Putin attended a ceremony in the Kremlin on December 12 to award medals to "Heroes of Labor" and state medals in the fields of science, technology, literature and the arts, the TASS news agency, Komsomolskaya Pravda and many other Russian media reported. He praised the Russian movie "Challenge" at the ceremony, calling the movie shot in space "a breakthrough in the global film industry. According to a Kremlin release… |  | *Falsified* |
| **#2:** A missile attack on a residential building in a central Ukrainian city has left 3 dead and 25 injured |  | Outstanding Women In Ukrainian History… Women are gracefulStock Photos, Royalty... Young girls in ethnic clothes walking in fields. Fashion photo, folklore style… |  | *Falsified* |
| **#3:** Tandesse says many countries understandably impose restrictive measures on Chinese inbound travelers, |  | On December 30, Foreign Ministry spokesman Wang Wenbin chaired a regular press conference. A reporter asked, WHO Director General Tandace said that due to the lack of comprehensive information from China, it is understandable that countries impose restrictive measures on travelers entering China in a way that they believe can protect their own populations. What does the spokesperson have to say about this…. |  | *Pristine* |
| **#4:** A late-night rollover of a wedding bus in Australia has left 10 dead and 25 injured. |  | At least 10 people were killed and 25 others were injured when a bus carrying wedding guests rolled down a slope at a roundabout in Australia's New South Wales state on June 11, local time, Reuters reported. The cause of the accident is under investigation. According to reports, the accident occurred near the town of Greta in the Hunter Valley, located about 180 kilometers northwest of Sydney, at about 11:30 p.m. local time that night. … |  | *Pristine* |

**Figure 8: Examples of Synthetic Chinese Multi-modal Fact-Checking dataset (Chinese is translated into English).**

| Claim Text | Claim Image | Text Evidence | Image Evidence | Truthfulness |
|---|---|---|---|---|
| **#1:** Three Gorges Dam release makes flooding in areas downstream of the dam "worse" |  | July 27, the Yangtze River Three Gorges Hub Project opened the flood relief deep hole flood discharge. August 4, Typhoon "Hegebi" landed in Zhejiang, by the impact of the typhoon, Wenzhou, serious flooding. Although many places have set a "small goal" to ensure that 2020 to eliminate urban flooding, however, 2020 since the beginning of the flood, from Guangzhou to Tianjin, from Wenzhou to Chongqing, "the city to see the sea" difficult to go. Our reporter checked…. |  | *Falsified* |
| **#2:** Dengue fever is airborne |  | Now with the heat and rainstorms in some places mosquitoes arrogant raging in many places have sounded the alarm of dengue fever prevention and at the same time, about the dengue fever rumors have begun to "stupid", triggering public concern. Here, "Zhen Zhen" and you explore those things about dengue fever - 1. Dengue fever can be spread through the air? Dengue fever is not airborne. Dengue fever is…. |  | *Falsified* |
| **#3:** Sunny rose grapes sicken in many parts of Japan, government studies countermeasures |  | According to Japan's Kyodo News Agency reported on the 24th, Japan's Ministry of Agriculture, Forestry and Fisheries recently implemented a questionnaire survey, planting sunshine rose grapes in 46 prefectures, there are 30 areas of grapes, "non-flowering disease". The Japanese government is conducting an urgent study, and strive to come up with countermeasures before the problem becomes serious. Kyodo News Agency said… |  | *Pristine* |
| **#4:** Supercharged DNA Repair Keeps Bowhead Whales Safe From Cancer |  | New Scientist website reported on the 22nd, bowhead whales are the world's longest-living mammals, rarely affected by cancer. U.S. scientists found in a new study, bowhead whale cells seem to be able to repair DNA more quickly and efficiently than human or mouse cells, which may explain why they can live to more than 200 years old and have a lower incidence of cancer. In the latest study, University of Rochester scientists…. |  | *Pristine* |

**Figure 9: Examples of Collected Chinese Multi-modal Fact-Checking dataset (Chinese is translated into English).**

dataset, we collect pristine claims and their corresponding documents from official websites such as 'Xinwen', 'Xinhua', 'Huanqiu', and others.

## C.2 Visualization of Matching Methods

In Figure 7, we present four instances of falsified claims from our constructed SCMF dataset:

- By searching for the person entity 'Trump', we retrieve unrelated news event images.
- Searching for the location entity 'Ukrainian' leads us to the image of Ukrainian women that is entirely unrelated to the original text.
- Searching for the organization entity 'police station' yields an image of a police station not located in Myanmar.
- The image is randomly matched with the text; we can observe that the image 'Putin' is entirely unrelated to the mention of 'Kherson Region' in the text.

In order to prevent the retrieval of images that remain related to the claim text, we adopt the following technique: During the image retrieval process using the Baidu Search API [2], we retain the caption or sentence from the link associated with the bottom of the image, which often describes the image's content. Subsequently, we leverage the state-of-the-art sentence-transformers, specifically all-MiniLM-L6-v2 [3], to compute the semantic similarity between the retrieved sentence and the claim text. By setting a threshold of 0.5, we retain text-image pairs with similarity scores below this threshold to acquire fabricated claims.

## C.3 Case Study

We show some data cases from both the CCMF and SCMF datasets in Figure 8 and Figure 9. We can observe that: (1) In the CCMF dataset, the falsified claims come from rumors on Internet platforms, and their claim text is often fabricated. Conversely, in the SCMF dataset, falsified claims have both claim text and claim images derived from real news, albeit incorrectly matched. (2) Evidence documents in the CCMF dataset are often sourced from fact-checking websites and contain discerning statements. In contrast, evidence documents in the SCMF dataset are sourced from textual content returned by Google API searches on images.

## D PARAMETER ANALYSIS

In our ESCNet, the backbone parameters used to extract the basic features are frozen and we only train the Strance Transformer, Entity-enhanced Encoder and Classifier. As shown in Figure 10, in comparison to several other models, our ESCNet significantly reduces the number of parameters while maintaining a leading performance, affirming the efficiency of our model design. This suggests that:

- The ESCNet jointly models both stance semantic reasoning features and knowledge-enhanced entity pair features, facilitating the learning of effective semantic-level and knowledge-level claim representations.

---

[2]https://image.baidu.com/
[3]https://huggingface.co/sentence-transformers/all-MiniLM-L6-v2

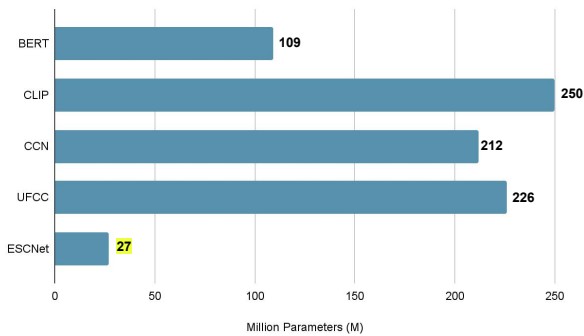

Figure 10: Comparison of the number of parameters between ESCNet and other models.

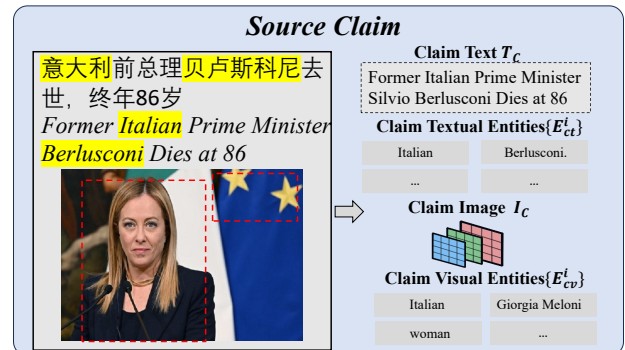

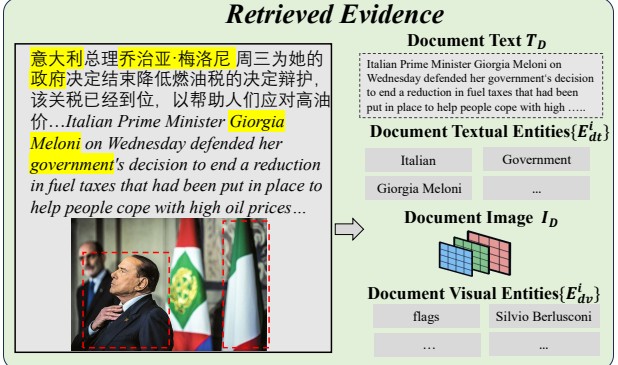

Figure 11: Visualization of extracting multi-modal cues in Multi-modal Feature Extraction Module.

- The Stance Transformer adeptly discerns the stance of evidence in relation to the claim across varying modalities.
- The Entity-enhanced Encoder connects entities from texts and images to the Knowledge Graph (KG), thereby acquiring high-level semantic insights and enabling fact-checking at the knowledge level.

## E    MORE RELATED WORK

### E.1    Fact-checking Models on Text.

Fact-checking in texts has always received widespread attention: and researchers regard it as a kind of Recognizing Textual Entailment (RTE) task [17], where the goal is to predict whether the text proves or disproves the claim. FEVER [41] was a fact verification method that employs bidirectional long short-term memory networks (Bi-LSTM) to encode claims and evidence separately; DeClarE model used a convolutional neural network (CNN) and attention mechanism to process text data, capturing the correlation between claims and evidence [32]; BERT-based fact-checking methods leveraged the pre-trained BERT (Bidirectional Encoder Representations from Transformers) model to provide powerful text representations for fact-cheking tasks; Zhang *et al.* [47] developed a model that utilizes dual emotion features to detect fake news online. They found that both publisher emotion and social emotion played significant roles in distinguishing fake news from real news. Cheng *et al.* [10] developed VRoC, a tweet-level variational autoencoder-based rumor classification system, to address the negative impacts of rumors spread through social media. And there were also many fact-cheking models that utilize graph neural networks [27, 28].

### E.2    Multi-modal Fact-checking Models.

While most existing research primarily focus on analyzing text, there are initial attempts to explore the integration of multi-modal information [8]. Khattar *et al.* [22] developed the Multimodal Variational Autoencoder (MVAE), an end-to-end network for fake news detection. MVAE combined a bimodal variational autoencoder with a binary classifier to learn shared representations of textual and visual information. Zhang *et al.* [48] used a structure coherence-based approach with components such as textual feature similarity, textual semantic similarity, text length and image similarity. Sahar *et al.* [1] proposed the consistency-checking network (CCN), which mimicked layered human reasoning across the same and different modalities and utilized diverse multimodal clues. Yao *et al.* [44] adopted the ensemble method by using different pre-trained models and several co-attention modules. Yao *et al.* [46] used the CLIP encoder and adopted a stance detection framework. Dhankar *et al.* [13] used a straightforward approach that concatenated the claim and documented textual (visual) representations and their cosine similarity. Du *et al.* [14] proposed a model with pre-trained DeBERTa for text and Swinv2 for image embeddings, that are combined using a co-attention fusion block. Gao *et al.* [39] experimented with a inter-modality and intra-modality fusion of textual and visual embeddings using the co-attention mechanism for their classification model and refered to this architecture as Multimodal Attention and Fusion Network (MAFN). Zhuang *et al.* [49] integrated disturbance on the embedding layer, a new loss function, and data augmentation by sequential dropout layers into the vanilla RoBERTa. Lee *et al.* [25] proposed a unifying textual and visual matching layer to confuse the two modality information. Gao *et al.* [16] proposed an ensemble model architecture by extracting various information for each modality individually. They applied multiple attention mechanisms to learn the multimodal interaction between visual and textual content pairs.

**Dataset.** A single example in the dataset consist of the following:

- A claim **image** *I*.
- A claim **text** *T*.
- **Visual evidence:**
  - A list of **images:** *I* = [*I*, ..., *I*].
- **Textual evidence:**
  - A list of **entities:** *ENT* = [*E*, ..., *E*].
  - A list of **sentences:** *S* = [*S*, ..., *S*].

**Task.** Classify {*I*, *T*} to: *Pristine or Falsified*.

**Figure 12: NewsCLIPpings dataset.**

## F    MORE QUALITATIVE EXPERIMENTS

As shown in Figure 11, we show the process of extracting multi-modal cues in the Multi-modal Feature Extraction Module: Regarding text entities extraction, the named entity linking tools bert-base-chinese-ner [35] and Tagme are applied to link the ambiguous entity to their corresponding entities in Freebase [5]. Regarding image entity extraction, due to the high precision required for pre-trained models, we exploit the APIs from the Baidu OpenAI platform to identify the objects and celebrities from the images.

## G    NEWSCLIPPINGS DATASET

Luo *et al.* [29] proposed a method that automatically, yet nontrivially, matches images accompanying real news with other real news captions. They used trained language and vision models to retrieve a close and convincing image given a caption. While this work contributes to misinformation detection research by automatically creating datasets, but it also amplifies the risk of generating falsified data on a large scale. The dataset [1] use the NewsCLIPpings [29] that contains both pristine and falsified ('out-of-context') images. It is built on the VisualNews [26] corpus that contains news pieces from 4 news outlets: The Guardian, BBC, USA Today, and The Washington Post. The NewsCLIPpings dataset contains different subsets depending on the method used to match the images with captions (e.g., text-text similarity, image-image similarity, etc.). We use the 'balanced' subset that has representatives of all matching methods and consists of 71,072 train, 7,024 validation, and 7,264 test examples. The NewsCLIPpings dataset components and task are summarized as Figure 12. Our model is applicable to evidence that contains multiple images, but for simplification, we assume that only a single image is present in a piece of evidence.

## H    DISCUSSION AND LIMITATION

Overall, we have proposed a multi-modal fact-checking framework that achieves state-of-the-art performance on three datasets. However, this task still faces numerous challenges, and relying solely on automatic fact-checking tools can have dangerous consequences: on one hand, incorrectly labeling original posts as rumours can negatively affect the spread of digital content, potentially impacting the revenue-generating capabilities of individuals or organizations

disseminating information. On the other hand, the adverse effects on social stability are even greater when rumour-laden posts are mistakenly labeled as truthful due to uncontrolled dissemination. Through the analysis of failed cases, we discover an interesting phenomenon: when the text or image evidence provided in the dataset is missing, irrelevant, or even mislabeled, ESCNet may not be able to make the correct judgment of the claim through the learned parameters. This also shows that this task still faces many challenges, and relying solely on automated fact-checking tools can have dangerous consequences. This motivates us to solve such problems in future work.

