# OpenReview forum: "ESCNet: Entity-enhanced and Stance Checking Network for Multi-modal Fact-Checking"
_ACM.org/TheWebConf/2024/Conference — TheWebConf24_

### Official Review · Reviewer_Xcxi · 2023-11-21

**Novelty:** 6
**Technical Quality:** 7

**Review:**

In this paper teh authors present the first large-scale, multi-domain Chinese multimodal fact-checking dataset. The authors also describe a Entity-enhanced Encoder with a
knowledge enhanced distance measurement strategy and a signed attention
mechanism to capture high-level entity information.

The paper is interesting, although a little long (with appendices the paper is over 15 pages). In general, the paper is well written, although the presentation could be improved. Some parts could perhaps be omitted (e.g. I think Floyd Warshall's algorithm is quite well known and does not need to be mentioned in the paper). Overall, I think it is a good paper that deserves to be published. The main contribution is the presentation of a new dataset that will be of interest to researchers interested in natural language processing and the web. I would like to suggest that the authors expand the section on the construction of the dataset. Indeed, they make some choices in the pre-processing phase that have a crucial impact on the final quality of the dataset, and it is appropriate that these choices are deeply motivated in order to understand what are the limitations of the use of the dataset. It would be interesting to understand whether the procedure used by the authors can be reused to produce datasets in languages other than Chinese and, if so, what effort is required to produce such an adaptation. Overall, a good paper, well written (although, I repeat, the presentation can be improved, which is of interest to the community).

**Questions:**

Please detail better the procedure for constructing the dataset

**Reviewer Confidence:**

2: The reviewer is willing to defend the evaluation, but it is likely that the reviewer did not understand parts of the paper

**Scope:**

4: The work is relevant to the Web and to the track, and is of broad interest to the community

---

### Official Review · Reviewer_TrUH · 2023-11-22

**Novelty:** 5
**Technical Quality:** 6

**Review:**

This research tackles the pressing issue of misinformation, highlighting the potency of multi-modal content (text and images) in social media dissemination. While previous research makes strides in feature extraction, it falls short in fully harnessing semantic and entity information. Additionally, existing datasets primarily cater to English and single types of misinformation, resulting in incomplete coverage.
In response, the authors construct the first large-scale Collected Chinese Multi-modal Fact-Checking (CCMF) and Synthetic Chinese Multimodal Fact-Checking (SCMF) benchmarks for evidence-based, multi-type, multi-modal fact-checking, encompassing a wide array of misinformation types. They introduce ESCNet, a novel model amalgamating semantic reasoning and knowledge-enhanced features, which includes Multi-modal Feature Extraction Module, Stance Transformer, and Entity-enhanced Encoder. ESCNet not only surpasses previous models but also sets a new standard for evidence-based, multi-modal fact-checking.

a.Strengths:

S1: The authors construct the first large-scale Collected Chinese Multi-modal Fact-Checking (CCMF) and Synthetic Chinese Multimodal Fact-Checking (SCMF) benchmarks.

S2: The experimental results illustrate the effectiveness and robustness of the proposed ESCNet framework, incorporating a wide range of sota baselines for comparative analysis.

S3: Clear and well-motivated reasoning in the paper. Well-written and structured.

b.Weaknesses:

W1: It's best to be able to provide the code for checking.

W2: I suggest to add the percentage of improvement in Figure and Table compared with other baselines to help read.

W3: The results of the model are extremely dependent on Multi-modal Feature Extraction Module. If the extraction effect is not good, it is extremely easy to accumulate errors. How to deal with this problem? I suggest to combine the overall textual and visual feature and perform ablation study to verify the Multi-modal Feature Extraction Module.

**Questions:**

Q1: How to deal with this problem stated in W3?  I suggest to combine the overall textual and visual feature and perform ablation study to verify the Multi-modal Feature Extraction Module.

Q2: Will the CCMF and SCMF datasets and ESCNet Model be open sourced?

Q3: Previous study mainly focused on Graph Pool to generate the overall representation using Graph Neural Networks(GNNs), what is the difference between  Entity-enhanced Encoder and GNNs Encoder such as GCN and GAT?

(I would be glad to change my score if the code and data will be open sourced)

**Reviewer Confidence:**

4: The reviewer is certain that the evaluation is correct and very familiar with the relevant literature

**Scope:**

4: The work is relevant to the Web and to the track, and is of broad interest to the community

---

### Official Review · Reviewer_SKAi · 2023-11-22

**Novelty:** 4
**Technical Quality:** 4

**Review:**

The article introduces the Entity-enhanced Stance Checking Network (ESCNet), a novel approach to multimodal fact-checking. This model is significant for its integration of multi-modal feature extraction, stance transformation, and an entity-enhanced encoder. The authors also present the first large-scale Chinese Multi-modal Fact-Checking (CMFC) dataset, encompassing 46,000 claims. This dataset is noteworthy for covering all types of misinformation and being divided into two sub-datasets: Collected Chinese Multi-modal Fact-Checking (CCMF) and Synthetic Chinese Multi-modal Fact-Checking (SCMF). In addition, The author compared many novel Fact-Checking methods in the experimental section and achieved SOTA results on the NewsCLIPpings dataset, which havecertain application values. However, there are some weaknesses that need to be further improved.

1. Dataset Specificity: The focus on Chinese-language data, while valuable, might limit the generalizability of the findings and the applicability of the model in other linguistic contexts.

2. Technical Details: The paper could benefit from more detailed explanations of the technical aspects of the ESCNet, particularly how it integrates and processes multimodal data.

**Questions:**

1. How applicable is the ESCNet model to languages and datasets other than Chinese? Are there specific modifications needed for such adaptations?
2. Can you elaborate on the technical workings of the entity-enhanced encoder and how it integrates with the stance transformer to enhance the fact-checking process?
3. What are the potential future directions for this research, particularly in terms of expanding the dataset to other languages and improving the model's capabilities?

**Reviewer Confidence:**

3: The reviewer is confident but not certain that the evaluation is correct

**Scope:**

3: The work is somewhat relevant to the Web and to the track, and is of narrow interest to a sub-community

---

### Official Review · Reviewer_Jhzw · 2023-11-27

**Novelty:** 5
**Technical Quality:** 6

**Review:**

The paper presents a good contribution in multi-modal fact-checking with the first large-scale Chinese Multi-modal Fact-Checking (CMFC) dataset. It comprises 46,000 claims, covering diverse types of misinformation. This dataset includes both Collected Chinese Multi-modal Fact-Checking (CCMF) and Synthetic Chinese Multi-modal Fact-Checking (SCMF) sub-datasets.

To benchmark performance, the paper introduces the Entity-enhanced and Stance Checking Network (ESCNet), a novel model incorporating a Multi-modal Feature Extraction Module, Stance Transformer, and Entity-enhanced Encoder. ESCNet can jointly modeling semantic reasoning features and knowledge-enhanced entity pair features.

The combination of data building and model innovation makes this work valuable and impactful in advancing the field of evidence-based, multi-type, and multi-modal fact-checking.

Pros:  Great efforts of a new dataset.

Cons:
- The writing of the model ESCNet is overall lacking a lot of details (perhaps due to space limits.)
- The improvement of ESCNet in Table 3, 4, 5 seems convincing, but it might be due to more parameters and complexity of the model.

**Questions:**

Are you going to make the dataset publically available? Have you considered creating a parallel English dataset, considering that Chinese-to-English translation is relatively straightforward? This would enhance the accessibility and utility of your research for other researchers in the field.

**Reviewer Confidence:**

3: The reviewer is confident but not certain that the evaluation is correct

**Scope:**

4: The work is relevant to the Web and to the track, and is of broad interest to the community

---

### Official Review · Reviewer_gHHR · 2023-12-01

**Novelty:** 5
**Technical Quality:** 4

**Review:**

Authors propose ESCNet (Entity enhanced and Stance Checking Network) for multi-modal fact checking. ESCNet consists of three
parts: Multi-modal Feature Extraction Module, Stance Transformer and Entity-enhanced Encoder. Authors also build a large-scale multi-modal fact checking dataset with ground truth.

Strength:
1. A very large-scale dataset is generated, which is publicly available could benefit this research.

2. Experimental results show that the proposed method is better than the competing methods by health margin.

3. Ablation study is provided to show the contribution of various components of the model.

Weakness:
1. The writing of the paper is poor. Authors discussed different components of the model without providing much justification of the role of the different components, which make the paper hard to read and follow.

2. it is not clear how the knowledge-enhanced distance measurement works and how it helps.

3. Only one public dataset is used to validate.

**Questions:**

Please respond to the weakness comments.

**Ethics Review Description:**

No issue

**Reviewer Confidence:**

2: The reviewer is willing to defend the evaluation, but it is likely that the reviewer did not understand parts of the paper

**Scope:**

3: The work is somewhat relevant to the Web and to the track, and is of narrow interest to a sub-community

---

### Decision · Program_Chairs · 2024-01-22

**Decision:**

Accept

**Comment:**

All reviewers gave positive scores to this paper.
 They point out the technical contributions, while some reviewers also criticize certain aspects of writing that should be improved.
 Please take their detailed comments into account when preparing the next version.